# The P2X7 Receptor in Oncogenesis and Metastatic Dissemination: New Insights on Vesicular Release and Adenosinergic Crosstalk

**DOI:** 10.3390/ijms241813906

**Published:** 2023-09-09

**Authors:** Elena Adinolfi, Elena De Marchi, Marianna Grignolo, Bartosz Szymczak, Anna Pegoraro

**Affiliations:** 1Section of Experimental Medicine, Department of Medical Sciences, University of Ferrara, 44121 Ferrara, Italy; elena.demarchi@unife.it (E.D.M.); marianna.grignolo@unife.it (M.G.); anna.pegoraro@unife.it (A.P.); 2Department of Biochemistry, Faculty of Biological and Veterinary Sciences, Nicolaus Copernicus University in Torun, 87-100 Torun, Poland; b_szymczak@doktorant.umk.pl

**Keywords:** P2X7, ATP, cancer, metastasis, exosomes, microvesicles

## Abstract

The tumor niche is an environment rich in extracellular ATP (eATP) where purinergic receptors have essential roles in different cell subtypes, including cancer, immune, and stromal cells. Here, we give an overview of recent discoveries regarding the role of probably the best-characterized purinergic receptor in the tumor microenvironment: P2X7. We cover the activities of the P2X7 receptor and its human splice variants in solid and liquid cancer proliferation, dissemination, and crosstalk with immune and endothelial cells. Particular attention is paid to the P2X7-dependent release of microvesicles and exosomes, their content, including ATP and miRNAs, and, in general, P2X7-activated mechanisms favoring metastatic spread and niche conditioning. Moreover, the emerging role of P2X7 in influencing the adenosinergic axis, formed by the ectonucleotidases CD39 and CD73 and the adenosine receptor A2A in cancer, is analyzed. Finally, we cover how antitumor therapy responses can be influenced by or can change P2X7 expression and function. This converging evidence suggests that P2X7 is an attractive therapeutic target for oncological conditions.

## 1. Extracellular ATP and Its Receptors in the Tumor Microenvironment

Extracellular ATP (eATP) is an established tumor microenvironment (TME) component that can be released following tumor and adjacent tissue necrosis via active release, mediated by proteins such as those of the ABC cassette family, or inside vesicles released from the tumor, immune, and stroma cells [1,2] (Figure 1). The live measurement of eATP in murine models was made possible by the development of the luciferase-based probe pmeLUC that, thanks to its specific placement at the extracellular facet of the plasma membrane, allows for distinguishing it from intracellular ATP [3,4,5]. Several studies, taking advantage of either pmeLUC or similar probes, have demonstrated eATP abundance in the TME, and that different treatments, including chemotherapy and caloric restriction, can affect its levels [6,7,8,9]. These data led Kamata-Sakurai and colleagues to develop a CD137 targeting antibody endowed with an ATP binding domain allowing for its specific release only at tumor sites, thus preventing systemic drug toxicity [10]. eATP is the natural ligand of two families of purinergic receptors: ionotropic P2Xs and metabotropic P2Ys [11]. Both classes of proteins are expressed at different levels by the cells of the TME and activate signaling pathways favoring growth, invasion, angiogenesis, and chemotherapy resistance, but also regulate, in a tumor-promoting or tumor-eradicating fashion, the immune system [1,2,12,13]. This overview focuses on the role of probably the most studied ATP receptor in cancer: P2X7. Converging evidence supports the view that P2X7 is the receptor for eATP most heavily involved in tumor–host interactions [14]. In the TME, ligands and receptors are involved in a positive feedback loop as eATP ligates P2X7 and triggers P2X7-mediated responses, among which is release of ATP; therefore, P2X7 can upregulate the concentration of its agonist [15]. Here, we cover the involvement of P2X7 in cancer growth, neovascularization, interactions with the immune system, and metastasis, concentrating on recent discoveries related to the release of miRNA-containing vesicles and the crosstalk among P2X7/CD39/CD73 and A2A receptors.

## 2. The P2X7 Receptor and Its Splice Variants

The P2X7 receptor is a low-affinity ATP-gated channel mediating, upon ligand engagement, the cellular influx of sodium and calcium and the efflux of potassium ions. P2X7-dependent potassium flux is associated with probably the best-known function of the receptor: activation of the NLRP3 inflammasome followed by maturation and secretion of the pro-inflammatory cytokines IL-1β and IL-18 [16,17,18]. Prolonged stimulation of the P2X7 receptor with high concentrations (mM range) of ATP also mediates the opening of large non-selective macropores permeable to solutes, such as ethidium, propidium, and lucifer yellow [19,20]. Macropore opening is associated with cell death by necrosis, apoptosis, or pyroptosis, depending upon the involved cell type [17,21]. However, basal stimulation of the receptor with subthreshold concentrations of the agonist, such as those in the TME, was also associated with trophic properties [1,22]. The functional P2X7 receptor is a homotrimer [23] formed by subunits of 595 amino acids whose tridimensional structure was recently identified and exploited for identifying and designing new receptor ligands [24,25,26,27,28,29,30]. The P2X7 subunit is formed by a short N terminal domain, two transmembrane regions, a large extracellular domain with ligand binding sites, and a long intracellular C terminal tail. The C terminal domain is responsible for macropore formation [31] and interactions with several proteins [32,33]. In humans, the gene for P2X7 is located on chromosome 12 [34]; it is highly polymorphic and, upon alternative splicing, can give rise to several variants [35,36]. The fully functional receptor is termed P2X7A. Among the other splice variants, P2X7B, P2X7H, and P2X7J have also been associated with oncogenic conditions [37,38,39,40,41,42,43,44]. The P2X7B isoform gives rise to a functional ion channel that lacks the pore-forming and apoptotic activity associated with P2X7A [37,45]. The alternative splicing is due to the retention of an intron between exons 10 and 11 of the full-length receptor that determines the addition of 18 extra amino acids after residue 346, followed by a stop codon leading to truncation of the entire C tail [37]. Moreover, when associated with P2X7A in a heterotrimer, P2X7B is a potentiating subunit upregulating both the channel and macropore activity [39,45].

On the contrary, P2X7J does not give rise to a functional protein but behaves as an antagonizing subunit toward P2X7A [38]. P2X7H, also called P2RX7-V3, generates a long non-coding RNA endowed with tumor-promoting activities [40]. Clinical studies with a blocking antibody specifically recognizing the so-called non-functional P2X7 (nfP2X7) [46] have been carried out by the Biosceptre company, showing promising results both in human and feline patients [14,47]. Although, probably for propriety reasons, we do not know the exact nature of the nfP2X7 variant, we do know that it does not entirely lose the channel activity. At the same time, it is not functional as an ion macropore [46]. Recent studies and overviews have addressed the role of alternatively spliced P2X7 isoforms in health and disease—we refer the reader to the literature for further detail on their activity [35,36,48,49].

## 3. The P2X7 Receptor in Cancer Growth and Immune Responses

The notion that the P2X7 receptor, under particular circumstances (i.e., low ATP concentrations or certain cell types not able to form cytolytic pores), could also exert trophic activity dates back to approximately twenty years ago [22,50,51] when we demonstrated that the receptor upregulates mitochondrial and reticular calcium levels, leading to increased metabolic activity [22,52]. However, many scientists were sceptical about the ability of a cytotoxic receptor to facilitate cancer growth until in vivo proof clearly emerged [53]. Since then, many papers have confirmed P2X7-dependent cancer-promoting activity and associated the expression of the receptor with increased cell metabolism, neo-vascularization, and, in general, poor patient prognoses [1,14,54,55,56,57,58,59]. Several solid and liquid cancer types overexpress P2X7, for which P2X7-targeting drugs are potential therapeutic tools; these include acute myeloid and chronic lymphocytic leukemia [9,42,60,61,62,63], melanoma [43,64,65], glioma [66,67,68], neuroblastoma [41,44,55,69,70] prostate, breast, bone, and colorectal cancer [71,72,73,74,75,76,77]. As the above-cited literature demonstrate, P2X7 growth-promoting roles have been covered elsewhere; therefore, its extensive characterization is beyond the scope of the present overview. The wealth of observations on P2X7 in such a plethora of cancers suggest that the receptor acts as a positive regulator of tumor formation and evolution and, therefore, that its pharmacological blockade could be advantageous for oncological patients [14,78]. However, in the TME, P2X7 is expressed not only by cancer cells but also by many components of the innate and cell-mediated immunity, and it plays a role in both pro-inflammatory and tolerogenic responses [12,13]. Indeed, P2X7 activation increases inflammatory responses favoring cancer immune eradication but can even promote tolerance by causing TGF-β release from myeloid-derived suppressor cells (MDSCs) [79]. Hence, the impact of P2X7 on cancer growth versus tumor immune control is complicated. Indeed, we and others demonstrated that a lack of host P2X7, for example in null mice, could favor tumor growth by significantly reducing immune infiltrates and pro-inflammatory cytokines and increasing intratumoral Tregs and adenosinergic pathways related to immune suppression [15,80,81,82,83,84]. However, even in P2X7 null-hosts, when an implanted tumor expresses P2X7, the administration of P2X7 antagonists causes a significant reduction in neoplastic growth accompanied by a substantial increase in CD4^+^ infiltrates, and also a decrease in the expression of the fitness marker CD73 in Tregs [15]. On the contrary, in tumor models where the expression of P2X7 is mainly restricted to immune cells, a P2X7 positive allosteric modulator, if administered with anti-PD-1 molecules, could prove beneficial [85,86]. Moreover, specific cytokine profiles affected by P2X7 activation could be either antitumoral or tumor-promoting, depending on the TME context [8,87].

This picture is further complicated by the effects that, via P2X7, eATP exerts on different T cell populations, including effector and regulatory T cells, their ability to survive as memory cells, and exhaustion or senescence processes [12,13,88]. For example, two research groups proposed exploiting P2X7 activity in tumor-infiltrating cytotoxic T cells to improve adoptive cell therapy potential, but achieved opposing results [14,89]. Romagani and colleagues suggested that a lack of P2X7 could be beneficial to overcome tumor-infiltrating lymphocyte (TIL) senescence [14]. At the same time, Wanhaimer and collaborators proposed that in the same cellular and tumoral model, P2X7 plays a central role in maintaining the mitochondrial fitness of TILs and controlled activation of the receptor before injection can even increase the antitumoral activity of these lymphocytes [89]. Although these contrasting results could be partially reconciled by the different cytokine cocktails administered to TILs before reinjection [89], they suggest caution when considering the administration of P2X7-targeting compounds in therapeutic settings. In general, a personalized medicine approach with negative or positive allosteric modulators of P2X7, derived from preventive evaluation of the receptor in tumor samples, including cancer and immune cells, would be advisable from our point of view.

## 4. P2X7 and Metastasis

Metastatic cancer forms still represent one of the leading causes of tumor-related death. Metastasis formation is a multistep process that requires gaining a series of mutations or other acquired abilities from tumor cells that lead to dissemination, local invasion, systemic resistance to immune cells, pre-conditioning, and colonization of secondary sites, also known as the metastatic niche. The advancement of metastasis is strongly influenced by the ability of transformed cancer cells to co-opt the host immune system and transition to distinct states, a fact attributable to their stemness capabilities [90]. Among the first reports relating P2X7 to cancer pathogenesis are those that demonstrated the receptor favoring cancer cell migration and dissemination [91,92,93,94] and reported its influence on epithelial–mesenchymal transition (EMT) [95,96,97,98]. EMT is a process of cell transdifferentiation originally adopted during embryogenesis and co-opted during metastasis, causing an increase in cancer cell migration linked to loss of cell polarity and down-regulation of adhesion molecules [99]. This mechanism was confirmed as the basis of P2X7-mediated invasiveness in recent publications covering the negative prognostic role of the receptor in oncological conditions as different as neuroblastoma [44], triple-negative breast cancer [100] and muscle-invasive bladder carcinoma [101]. In addition to EMT, metastasis is favored by signaling pathways and mechanisms implicated in tissue regeneration, wound healing, and adaptation to stress [90] that have been associated with P2X7 activity both in cancer and other physio-pathological conditions [17,53,55,84,96,102,103,104,105,106,107,108]. Among those pathways activated by the P2X7 receptor, the HIF1α/VEGF axis is one of the best studied [53,84,109,110]. Moreover, P2X7 is associated with increased Myc expression in cancer [42,111], a condition promoting metastasis by increasing the invasion and survival of cancer cells in the bloodstream [112]. Finally, P2X7 promotes autophagy [17,44,106,107,108] and unfolded protein responses [113], which help confer metabolic and immune evasive plasticity in solid cancers [90]. Given the involvement of the P2X7 receptor in the aforementioned metastasis-promoting signaling pathways and mechanisms, it is not surprising that overexpression of the receptor was associated with metastatic stages of multiple cancers [43,55,101,114,115,116,117,118]. Accordingly, P2X7 antagonism proved efficacious in reducing the in vivo dissemination of cancer cells in animal models of metastasis [41,43,78,91,92,96,119].

Interestingly, an increasing amount of evidence tends to associate the metastatic properties of P2X7 with its isoform B. As previously mentioned, P2X7B is a splice variant, present only in humans, which retains the calcium channel properties of P2X7 but lacks the activity of the macropore [37]. We first associated the expression of this isoform with increased metabolic activities in HEK293 cells [45] and a stem-like proliferation-associated phenotype in osteosarcoma cells and patients [39]. Subsequently, Lameu and colleagues demonstrated that the metastatic properties conferred by bradykinin treatment to neuroblastoma cells could be reverted by blocking P2X7B [41]. The same authors later confirmed that in the absence of bradykinin, P2X7 isoform B was central in conferring a series of pro-metastatic features to neuroblastoma cells, including the acquisition of a stem-like phenotype, the suppression of autophagy and induction of EMT [44]. These data are in line with those obtained by Tattersall and colleagues who examined osteosarcoma murine models, where they demonstrated that P2X7B expression could reduce cell adhesion, promote invasion, and upregulate a genetic axis, including FN1/LOX/PDGFB/IGFBP3/BMP4 [75]. The same study also proved that P2X7B could increase osteosarcoma’s propensity to spread in mouse lungs, and that administration of its antagonist A740003 could abrogate cancer-associated ectopic bone formation [75]. More recently, P2X7B overexpression was also reported in cohorts of metastatic melanoma [43] and prostate cancer patients with bone metastases [76]. In this latest study, Wang and colleagues also suggested that P2X7 truncated variants can double bone skeletal tumors, thus strongly reducing survival in metastatic mice models via a bone tropism-related mechanism, including the production of IL-6 [76].

## 5. Role of the P2X7 Receptor in Cancer-Associated Vesicle Release

The release of vesicles from cells is an essential process in cell–cell communication during physiological and pathological processes. Extracellular vesicles (EVs) comprise double membranous particles of different sizes and composition. They can derive from the direct budding of cell membrane cell or the fusion of the intraluminal vesicles with the membrane. EVs can carry proteins, lipids, and nucleic acids, thereby affecting microenvironment composition and cell behavior [120]. Moreover, EVs released from cancer cells can promote pre-metastatic niche formation in organs distant from primary tumors [121,122,123]. Different stimuli can induce the release of vesicles and ATP through its receptors—P2X7 is one such receptor [124] (Figure 1). Activation of the receptor by ATP induces Ca^2+^ influx, eliciting the exocytosis of EVs in the extracellular space [125]. P2X7-dependent release of vesicles was initially demonstrated in immune cells such as macrophages [126,127], dendritic cells [128,129], and microglia [130] (Figure 1).

**Figure 1 ijms-24-13906-f001:**
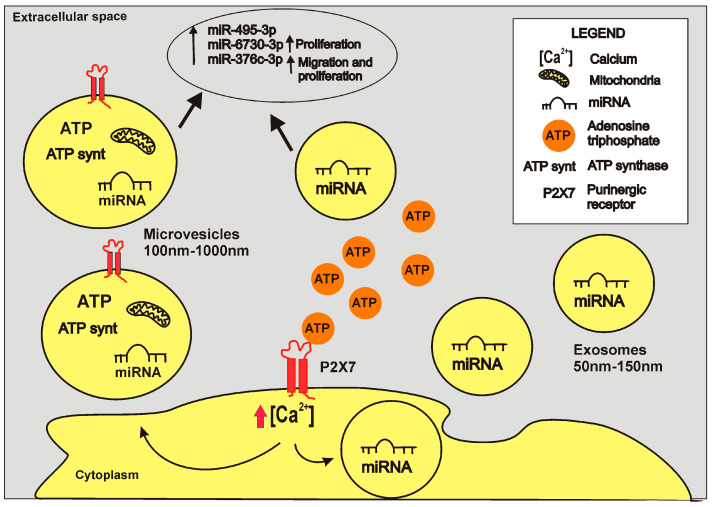
P2X7-dependent vesicle release. Upon ligation with extracellular ATP, P2X7 triggers the release of vesicles of different dimensions and natures from immune, cancer, neuronal, and glial cells. The figure represents vesicles released from tumor cells, including microvesicles/particles ranging in size from 100 to 1000 nm and containing ATP, miRNAs, mitochondria, and ATP synthase [43,58,131], and also exosomes ranging in size from 50 to 150 nm and containing miRNAs [43]. P2X7 activation increases the content of miRNAs, including miR-495-3p, miR-6730-3p, and miR-376c-3p in both vesicular fractions, while antagonism can block their release [43]. Some of these miRNAs have been shown to increase melanoma cells’ proliferation and migratory ability [43]. By carrying mitochondria and ectopic ATP synthase, microparticles can produce internal ATP [58,131]. Finally, vesicles also retain P2X7 on their surface, thus possibly facilitating the release of their content in areas rich in eATP.

These vesicles contain inflammatory cytokines and are central players in phlogistic reactions [129,132,133]. In recent years, several studies have focused on this cellular process, demonstrating the release of EVs following P2X7 stimulation from different cancer types, such as neuroblastoma [131,134], lung [131], prostate [135], breast cancers [131,136], and also melanoma [43,58]. EVs contribute to tumor progression, cell migration [43,136], and pre-metastatic niche conditioning [136]. P2X7 activation can induce the release of heterogeneous small and large vesicles [43,131], which different authors have identified as microvesicles, microparticles, exosomes, and oncosomes; therefore, exploring their contents can help to elucidate their biological effects. P2X7 stimulation can induce the release of vesicles containing ATP, thereby increasing its concentration in the TME [58]. ATP is loaded into vesicles thanks to the vesicular nucleotide transporter (VNUT) [137]. However, ATP can also be generated inside EVs in the extracellular environment thanks to glycolytic enzymes located inside the vesicles released from cancer cells [138]. In line with these data, analysis of the content of EVs released upon P2X7 activation demonstrated that they contain other essential players in ATP production: mitochondria [58] and the ectopic ATP synthase [131]. Vesicular ATP can also activate P2X7 in neighboring or distant cells, favoring a positive P2X7-dependent EV release loop [131,134,139]. Additionally, it can support several cellular processes involved in cancer progression, among which cell migration is fundamental for metastatic spread [124,140]. P2X7 can be delivered in vesicles [43,130,137] and, together with ATP, reach distant sites to potentially induce a pro-tumoral microenvironment and metastatic niche formation. The role of P2X7 expressed on the surface of EVs is not clear yet; however, it is possible to hypothesize that the receptor, via macropore formation, could trigger the release of EV content in eATP-rich areas, such as those associated with inflammation or the TME itself [58]. Together with ATP, other molecules, such as microRNAs (miRNA), are carried by EVs and can potentially modify cell behavior [141]. miRNAs are small non-coding RNA molecules that negatively regulate gene expression. Vesicle miRNA expression changes with cancer types and can promote diverse effects [141]. P2X7, through the interaction with the RNA-binding protein FMR1 can select the miRNA content of small EVs [142]. In line with these data, we demonstrated that microvesicles and exosomes released upon P2X7 stimulation contained a miRNA expression profile profoundly different from vesicles collected from unstimulated cells. Among the miRNAs upregulated in EVs released following P2X7 activation, miR-495-3p, miR-376c-3p, and miR-6730-3p showed proliferation and migration-promoting effects. Treatment of cells with the P2X7 antagonist A740003, before stimulation of EV release, reduced the expression of all three miRNAs [43], thus suggesting P2X7 antagonism as a possible pharmacological strategy to prevent pro-tumoral effects associated with vesicular release.

## 6. P2X7R and Its Crosstalk with the Adenosinergic Axis in Cancer

Recent data by us and other groups revealed an interplay between P2X7 and the adenosinergic axis formed by ectonucleotidases and adenosine receptors in the TME. This is unsurprising as eATP and its hydrolytic derivative adenosine are constituents of the TME [1]. As mentioned above, in the TME, eATP promotes tumor growth and immune-mediated tumor eradication, mainly via the P2X7 receptor [1]. Adenosine, generated from eATP via CD39 and CD73 ectonucleotidases, is an immune suppressant facilitating tumor escape, acting as an immune cell “don’t eat me” signal mainly via its activity at the A2A receptor (A2AR) [84,143,144,145]. However, the effects of both eATP and adenosine are not limited to immune cells, as often through the same receptors expressed by cancer cells, they can also promote cancer growth, vascularization, and metastasis [56] (Figure 2). To further complicate the picture, the effect of purines on immune cells is not always clearly pro- or antitumoral; for example, both ATP and adenosine are possibly required to produce the primary tumor-eradicating cytokine: IFN-γ [146]. Indeed, ATP and adenosine cooperatively stimulate the upregulation of the major histocompatibility complexes in dendritic cells, thus favoring the activation of T cells designated to IFN-γ secretion [125]. As mentioned above in the TME, the activity of the ectonucleotidases CD39 and CD73 closely controls eATP concentration. P2X7 can interfere with this process by modulating the expression of CD73 and CD39 in cancer-infiltrating immune cells and influencing the level of PD-1 in Tregs [15]. The overexpression of ectonucleotidases is one of the mechanisms causing a reduction in eATP in the TME in tumor-bearing P2X7 null mice [15]. The importance of the crosstalk between P2X7 and CD39 in the TME is further confirmed by the finding that CD39-targeting antibodies require a functional P2X7 receptor in immune cells to work appropriately as antitumoral agents in primary and metastatic murine tumor models [82,83]. These data are further corroborated by a recent study from Casey et al. demonstrating that P2X7 signaling was essential for the favorable effects of CD39 blockade in diffuse large B-cell lymphoma. Indeed, antagonism of CD39 caused an accumulation of eATP that acted on macrophage P2X7, favoring lymphoma cell phagocytosis. eATP-activated P2X7-dependent phagocytosis reinforced the activity of the lymphoma-targeting antibodies rituximab and daratumumab and facilitated therapy resistance [147].

**Figure 2 ijms-24-13906-f002:**
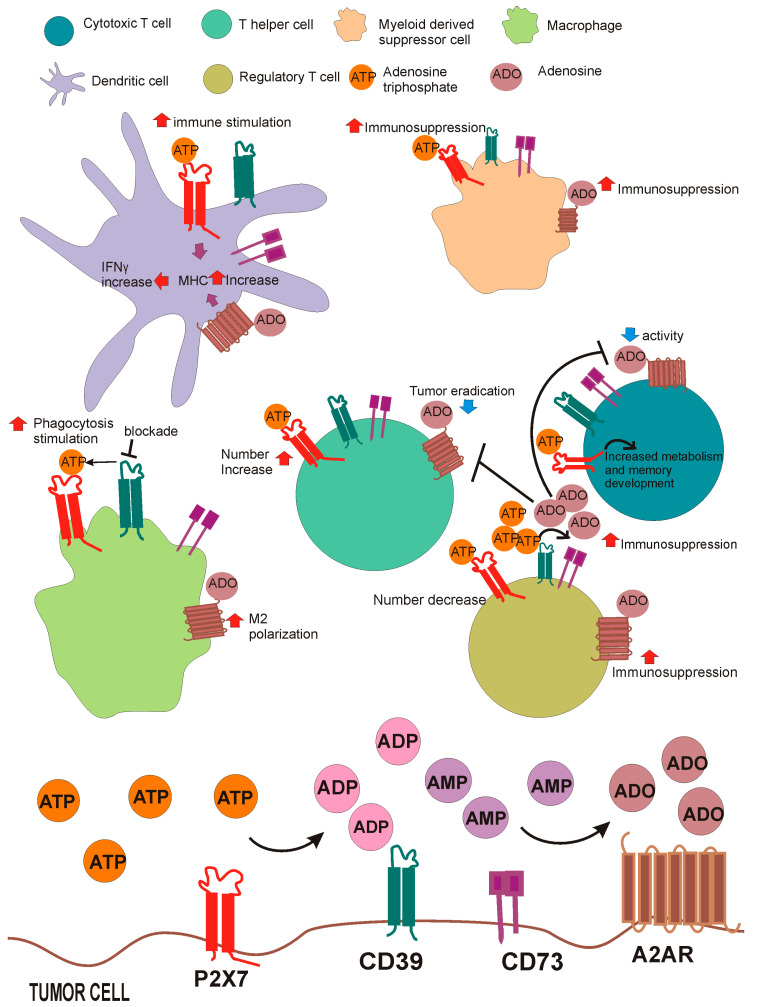
Molecules of the purinergic/adenosinergic axis are expressed by cancer and immune cells in the TME. The TME is rich in eATP due to necrosis and active release of the nucleotide. eATP acts on P2X7 and is then degraded by CD39 and CD73 to adenosine (ADO) acting as the primary source of this purine in the extracellular milieu [1,2]. The P2X7 receptor, the ectonucleotidases CD39 and CD73, and the A2A receptor are expressed by different immune cells involved in both tumor immune-eradication and suppression. In dendritic cells, eATP through P2X7 favors immune stimulation and MHC increase, which are also positively affected by ADO. In macrophages, adenosine sustains M2 polarization, while CD39 blockade, which causes eATP accumulation, stimulates phagocytosis via P2X7. Both eATP and ADO favor myeloid-derived suppressor cell (MDSC) activity [79,144]. ADO, produced in Tregs thanks to CD39/CD73, blocks the proliferation of effector CD4^+^ and CD8^+^ T cells via A2AR, contemporarily upregulating immune suppressive activity in Tregs. On the contrary, ATP via P2X7 reduces Treg numbers in tumor infiltrates [15]. P2X7 positively affects tumor infiltration via CD4^+^ T helper effector cells [15]. ADO via A2AR reduces cytotoxic cell activity [148,149], while P2X7 increases their metabolic activity, thus sustaining responses mediated by memory cells [88].

Recent data from our laboratory and others also point to an interaction between P2X7 and the A2AR in the TME. Indeed, tumors growing in P2X7 null mice overexpressed A2AR [84], possibly due to altered adenosine levels deriving from reduced eATP levels and increased ectonucleotidase activity in the TME [15]. Overexpression of A2AR is not limited to tumors but also affects spleen cells, leading to a systemic immune suppressive phenotype with a significant reduction in circulating IL-1β, TNF-α, IL-6, IL-12, IL-17, and IFN-γ levels, and an increase in TGF-β [84]. Interestingly, inside tumors, A2AR was incremented in necrotic areas, thereby increasing neovascularization, and its blockade caused a substantial reduction in VEGF production in tumor-bearing P2X7 null mice [84]. In a similar vein, a recent study reported upregulation of the P2X7 receptor in melanoma-bearing mice treated with the A2AR antagonist istradefylline [150].

Interestingly, this A2AR blocker increased P2X7 inside the tumors and in lymphoid organs and showed an effect on P2X7 expression even in the absence of cancer [133]. All the described evidence suggests that when developing anti-cancer treatments targeting single components of the P2X7/CD39/CD73/A2AR axis, scientists should consider their effects on the other members of the pathway that might reinforce or impair the efficacy of single protein-targeting drugs. New therapeutic approaches involving multiple targeting strategies could help overcome this issue.

## 7. P2X7 Receptor in Antitumoral Therapy Resistance

An increasing number of studies have associated P2X7 and its splice variants with either reinforcement or resistance to standard antitumoral therapeutic regimes [2]. eATP levels in the TME may be substantially increased by cytotoxic interventions, such as radio or chemotherapy, while certain antitumoral drugs, such as doxorubicin, daunorubicin, and oxaliplatin were even shown to induce extra release of ATP, which correlated with the onset of immunogenic cell-death-related tumor-eradicating immune responses [8,151]. In this context, opening P2X7A macropores in cancer cells can increase the efficacy of cytotoxic therapies, leading to better prognoses following treatments [42,44,67,152,153]. On the contrary, P2X7B, due to its different gating properties, may be positively selected by eATP and increased in the TME by these treatments, to sustain therapy resistance and relapse [42,44,152]. P2X7-dependent amplification of therapy efficiency was linked to the facilitation of daunorubicin intracellular load in AML blasts [42], increased cytotoxic activity of temozolomide in glioma cells [153], and enhanced differentiating efficacy of retinoic acid in neuroblastoma cells [44]. P2X7A overexpression is also a positive predictor of therapy responses in RAS-mutated melanoma patients, where it is associated with prolonged overall and progression-free survival and in general, early drug responses [65]. In this context, it is envisaged that the administration of positive allosteric modulators of P2X7A, in combination with standard therapy to reinforce their efficacy, can be followed by anti-P2X7B drugs to prevent resistance and relapse after classic therapy cycles [2,42,44,152].

## 8. Conclusions

Since the first publications proving the direct involvement of the P2X7 receptor in liquid and solid cancer growth [53,60,69], the number of studies analyzing its role in oncogenesis has increased tremendously, reaching more than 700 publications in PubMed (as of August 2023). Both the previous and recent literature strongly suggest that the P2X7 receptor is a suitable therapeutic target for oncological conditions, depending on the type of tumor, the splice variants of the receptor expressed, and the adenosinergic context. Old and recent literature strongly suggest the P2X7 receptor as a suitable therapeutic target in oncologic conditions. Depending on the type of tumor, the splice variants expressed and the adenosinergic context, P2X7 could be exploited to develop new therapeutic strategies based on its antagonism or agonism. Additionally, P2X7 targeting drugs could be or co-administered with traditional or innovative anti-cancer treatments to improve their efficacy. Finally, P2X7-blocking drugs could facilitate the development of novel therapeutic strategies to block the release of EVs from cancer or immune cells or alter their content to combat their metastasis-promoting activities.

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
