# Peer review of "The P2X7 Receptor in Oncogenesis and Metastatic Dissemination: New Insights on Vesicular Release and Adenosinergic Crosstalk"

_ijms, 2023, doi:10.3390/ijms241813906_

Round 1
Reviewer 1 Report
Comments for the review article “ Recent Advances on the Role of P2X7 Receptor in Oncogenesis 2 and Metastatic Dissemination”.
This is an interesting, well written and up to date article. Although other reviews have also focused on the role of P2X7 receptor in cancer, this manuscript has updated data and introduce two innovative topics to the discussion: the role of P2X7 receptor in release of small signaling vesicles and the crosstalk with adenosinergic signaling.
I have also some minor observation
1.- In line 25 the word “tumor” must be written in lower case.
2.-In line 66-67, an antagonist is also a ligand, the phrase should read: “a large extracellular domain with the ligand binding site”.
3.-In line 150, I think the more appropriated term to describe EMT is “transdifferentiation” not “transformation”.
4.-In line 173 change the word “man” by “humans”.
Author Response
This is an interesting, well written and up to date article. Although other reviews have also focused on the role of P2X7 receptor in cancer, this manuscript has updated data and introduce two innovative topics to the discussion: the role of P2X7 receptor in release of small signaling vesicles and the crosstalk with adenosinergic signaling.
We are grateful to the reviewer for appreciating our manuscript. We have now stressed that the focus of our overview is on vesicles and adenosinergic signaling by clearly stating it from the title that we updated into: "The P2X7 Receptor in Oncogenesis and Metastatic Dissemination: new insights on vesicular release and adenosinergic crosstalk."
I have also some minor observation
1.- In line 25 the word "tumor" must be written in lower case.
2.-In line 66-67, an antagonist is also a ligand, the phrase should read: "a large extracellular domain with the ligand binding site".
3.-In line 150, I think the more appropriated term to describe EMT is "transdifferentiation" not "transformation".
4.-In line 173 change the word "man" by "humans".
We are grateful to the reviewer for spending time checking our manuscript in detail. We apologize for the previous mistakes that have now been corrected according to the reviewer's suggestions.

Reviewer 2 Report
In the review manuscript titled 'Recent Advances on the Role of P2X7 Receptor in Oncogenesis and Metastatic Dissemination,' the authors summarize new findings on the P2X7 receptor, a well-known purinergic receptor in the tumor microenvironment. The review covers the functions of the P2X7 receptor and its human variants in both solid and liquid cancer growth and spread. It also explores how these receptors interact with immune and endothelial cells and focuses on the P2X7-dependent release of microvesicles and exosomes, along with their contents such as ATP and miRNAs. Additionally, the review analyzes the influence of P2X7 on the adenosinergic axis, involving ectonucleotidases CD39 and CD73 and the adenosine receptor A2A in cancer. Finally, it examines how anti-tumor therapies may affect or be affected by P2X7 expression and function. The evidence presented emphasizes the potential of P2X7 as a promising therapeutic target for cancer treatment.
Several review papers exist on this topic, so it is essential to highlight the innovation points and fresh perspectives offered by this paper.
The figures presented in this paper are difficult to read due to the lack of clear labels and the absence of necessary descriptions within the figures.
The recommended font types for the figures are Times New Roman or Arial.
Please remove the additional space in the manuscript.
Please check the spelling and formatting errors carefully.
Author Response
In the review manuscript titled 'Recent Advances on the Role of P2X7 Receptor in Oncogenesis and Metastatic Dissemination,' the authors summarize new findings on the P2X7 receptor, a well-known purinergic receptor in the tumor microenvironment. The review covers the functions of the P2X7 receptor and its human variants in both solid and liquid cancer growth and spread. It also explores how these receptors interact with immune and endothelial cells and focuses on the P2X7-dependent release of microvesicles and exosomes, along with their contents such as ATP and miRNAs. Additionally, the review analyzes the influence of P2X7 on the adenosinergic axis, involving ectonucleotidases CD39 and CD73 and the adenosine receptor A2A in cancer. Finally, it examines how anti-tumor therapies may affect or be affected by P2X7 expression and function. The evidence presented emphasizes the potential of P2X7 as a promising therapeutic target for cancer treatment.
Several review papers exist on this topic, so it is essential to highlight the innovation points and fresh perspectives offered by this paper.
We are grateful to the reviewer for the appreciation of our paper. We agree with the referee that the role of P2X7 in vesicular release from cancer cells and its interaction with adenosinergic signaling are our manuscript's main novelty and focus. Therefore, to further stress this point, we have now updated the title of the review to: "The P2X7 Receptor in Oncogenesis and Metastatic Dissemination: new insights on vesicular release and adenosinergic crosstalk".
New text sections and references were also included in the latest version of the manuscript.
The figures presented in this paper are difficult to read due to the lack of clear labels and the absence of necessary descriptions within the figures.
We apologize to the reviewer for the previous version of our figures. We have now substantially modified them, including legends, labels, and clear descriptions inside the pictures and extending figure legends.
The recommended font types for the figures are Times New Roman or Arial.
The figure font has been updated into Arial in their latest version.
Please remove the additional space in the manuscript.
We carefully checked the manuscript for extra spaces and fixed these mistakes.
Comments on the Quality of English Language
Please check the spelling and formatting errors carefully.
We thank the reviewer for raising this point. We have corrected spelling and typos all over the paper. If possible, we will also upload a new version of the manuscript with all changes highlighted in red.

Round 2
Reviewer 2 Report
Thank you to the author for addressing all of my concerns and comments. The work is now suitable for the journal.
Author Response
Thanks for giving me the opportunity of reviewing this manuscript. This is an interesting
state of art on P2X7 Receptor. The manuscript is well written, organized and, includes updated on
recent articles.
We thank the editor for her/his appreciation of our manuscript.
To my point of view, there is a major point which need to be corrected before acceptance for
publication.
The authors highlight the involvement of EVs in the role of P2X7 Receptor during metastasis.
According to the ISEV, now such EVs must be referred as small (50-150 nm) or large (100-1000 nm)
EVs. The authors use this classification within the text (lines 203-231) but not in the figure 1.
Distinctions between microvesicles and exosomes based on CLX and Alix detection is not used
anymore.
Figure 1 must be corrected according to small vs large EVs as well as effects from
microvesicles and exosomes must be combined.
We thank the editor for letting us notice this point. We have now corrected these mistakes and removed Alix and Calnexin from our figures.
A minor change is also requested in line 269. The authors use the word “stroma” which
indeed refers to fibroblasts, activated in CAF or not, and the extracellular matrix (mainly type I
collagen). Indeed, the figure 2 highlights only interactions of tumor cells with immune cells. Or
fibroblasts/CAF are in figure 2 or the “surrounding stroma” must be deleted from the line 269. If it is
retained, references are requested.
Thanks for your comment. The requested changes have now been applied.
Best regards,